# Acute Kidney Injury after Lung Transplantation: A Systematic Review and Meta-Analysis

**DOI:** 10.3390/jcm8101713

**Published:** 2019-10-17

**Authors:** Ploypin Lertjitbanjong, Charat Thongprayoon, Wisit Cheungpasitporn, Oisín A. O’Corragain, Narat Srivali, Tarun Bathini, Kanramon Watthanasuntorn, Narothama Reddy Aeddula, Sohail Abdul Salim, Patompong Ungprasert, Erin A. Gillaspie, Karn Wijarnpreecha, Michael A. Mao, Wisit Kaewput

**Affiliations:** 1Department of Internal Medicine, Bassett Medical Center, Cooperstown, NY 13326, USA; ploypinlert@gmail.com (P.L.); kanramon@gmail.com (K.W.); 2Division of Nephrology and Hypertension, Mayo Clinic, Rochester, MN 55905, USA; charat.thongprayoon@gmail.com; 3Division of Nephrology, Department of Medicine, University of Mississippi Medical Center, Jackson, MS 39216, USA; sohail3553@gmail.com; 4Department of Thoracic Medicine and Surgery, Temple University Hospital, Philadelphia, PA 19140, USA; 109426469@umail.ucc.ie; 5Department of Internal Medicine, St. Agnes Hospital, Baltimore, MD 21229, USA; nsrivali@gmail.com; 6Department of Internal Medicine, University of Arizona, Tucson, AZ 85721, USA; tarunjacobb@gmail.com; 7Department of Medicine, Deaconess Health System, Evansville, IN 47747, USA; dr.anreddy@gmail.com; 8Cleveland Clinic Lerner College of Medicine of Case Western Reserve University, Cleveland Clinic, Cleveland, OH 44195, USA; p.ungprasert@gmail.com; 9Department of Thoracic Surgery, Vanderbilt University Medical Center, Nashville, TN 37212, USA; erin.a.gillaspie@vumc.org; 10Department of Medicine, Mayo Clinic, Jacksonville, FL 32224, USA; dr.karn.wi@gmail.com (K.W.); mao.michael@mayo.edu (M.A.M.); 11Department of Military and Community Medicine, Phramongkutklao College of Medicine, Bangkok 10400, Thailand

**Keywords:** acute kidney injury, incidence, lung transplantation, transplantation, epidemiology, meta-analysis

## Abstract

Background: Lung transplantation has been increasingly performed worldwide and is considered an effective therapy for patients with various causes of end-stage lung diseases. We performed a systematic review to assess the incidence and impact of acute kidney injury (AKI) and severe AKI requiring renal replacement therapy (RRT) in patients after lung transplantation. Methods: A literature search was conducted utilizing Ovid MEDLINE, EMBASE, and Cochrane Database from inception through June 2019. We included studies that evaluated the incidence of AKI, severe AKI requiring RRT, and mortality risk of AKI among patients after lung transplantation. Pooled incidence and odds ratios (ORs) with 95% confidence interval (CI) were obtained using random-effects meta-analysis. The protocol for this meta-analysis is registered with PROSPERO (International Prospective Register of Systematic Reviews; no. CRD42019134095). Results: A total of 26 cohort studies with a total of 40,592 patients after lung transplantation were enrolled. Overall, the pooled estimated incidence rates of AKI (by standard AKI definitions) and severe AKI requiring RRT following lung transplantation were 52.5% (95% CI: 45.8–59.1%) and 9.3% (95% CI: 7.6–11.4%). Meta-regression analysis demonstrated that the year of study did not significantly affect the incidence of AKI (*p* = 0.22) and severe AKI requiring RRT (*p* = 0.68). The pooled ORs of in-hospital mortality in patients after lung transplantation with AKI and severe AKI requiring RRT were 2.75 (95% CI, 1.18–6.41) and 10.89 (95% CI, 5.03–23.58). At five years, the pooled ORs of mortality among patients after lung transplantation with AKI and severe AKI requiring RRT were 1.47 (95% CI, 1.11–1.94) and 4.79 (95% CI, 3.58–6.40), respectively. Conclusion: The overall estimated incidence rates of AKI and severe AKI requiring RRT in patients after lung transplantation are 52.5% and 9.3%, respectively. Despite advances in therapy, the incidence of AKI in patients after lung transplantation does not seem to have decreased. In addition, AKI after lung transplantation is significantly associated with reduced short-term and long-term survival.

## 1. Introduction

Acute kidney injury (AKI) is a complex clinical syndrome characterized by a sharp reduction in the glomerular filtration rate (GFR) followed by elevated serum creatinine or oliguria, and is associated with various etiologies and pathophysiological pathways. AKI is a major global health problem with a steadily increasing incidence in recent years [1,2,3]. The global burden of AKI is 13.3 million cases per year and is associated with significant mortality, resulting in 1.4 million deaths per year [4,5,6]. Mortality rates from AKI range from 16% to 50% according to the stage and vary widely according to etiology and patient comorbidities [7,8]. Those who survive the AKI are at increased risk for hypertension and progressive chronic kidney disease (CKD), including end-stage kidney disease (ESKD) [9].

Since the first human lung transplant was performed in 1963, almost 55,000 lung transplantations have been performed worldwide, now with nearly 4600 lung transplantations performed annually [10]. Up to 68% of lung transplant recipients develop AKI, which has been associated with increased one-year mortality, length of hospital stay, higher resource utilization, and related health care burden [10,11,12,13,14,15,16,17,18,19,20,21,22]. Though survival following lung transplantation has improved over the past few decades, morbidity and mortality related to AKI after lung transplantation and resultant progressive CKD remain relatively high and is a cause for increasing concern [16,23,24,25]. The incidence of AKI following lung transplantation varies widely, estimated to be as high as two-thirds of recipients, with up to 5% to 8% requiring dialysis in the initial few months post lung transplantation [11,13,14,15,21,24,26,27,28,29]. Differences in the definition of AKI may account for the variance of incidence of post-lung-transplant AKI [28].

Despite significant advances in lung transplantation surgical and medical practices, the epidemiology, risk factors, and mortality associated with AKI among post-lung-transplant recipients and their trends remain unclear. Therefore, we conducted a systematic review to summarize and trend the incidence of AKI (utilizing standard AKI definitions including AKIN (acute kidney injury network) [30], RIFLE (risk, injury, failure, loss of kidney function, and end-stage kidney disease) [31], and KDIGO (kidney disease: Improving global outcomes) [32] classifications) and mortality risk of AKI in lung transplant recipients.

## 2. Methods

### 2.1. Information Sources and Search Strategy 

The protocol for this meta-analysis is registered with PROSPERO (International Prospective Register of Systematic Reviews (CRD42019134095)). A systematic literature search of Ovid MEDLINE, EMBASE, and the Cochrane Database from database inceptions through June 2019 was performed to summarize the incidence and impact of AKI on mortality risk among adult patients following lung transplantation. Two investigators (P.L. and C.T.) individually performed a systematic literature search utilizing the search approach that consolidated the search terms of “lung” OR “pulmonary” AND “transplant” OR “transplantation” AND “acute kidney injury” OR “acute renal failure” OR “renal replacement therapy”. Detailed information on the search strategy from each database is provided in Online Appendix A. No language limitation was implemented. A manual review for conceivably-related studies employing references of the included studies was additionally performed. Grey literature was also searched for further relevant information. This systematic review was conducted following the PRISMA (Preferred Reporting Items for Systematic Reviews and Meta-Analysis) statement [33].

### 2.2. Study Selection

Studies were included in this meta-analysis if they were observational studies or clinical trials that provided data on incidence of AKI (utilizing standard AKI definitions including AKIN [30], RIFLE [31], and KDIGO [32] classifications), AKI requiring renal replacement therapy (RRT), and mortality risk of AKI in adult patients after lung transplantation (age ≥ 18 years old). Eligible studies needed to provide data to assess the incidence or mortality rate of AKI with 95% confidence intervals (CIs). Retrieved articles were individually examined for eligibility by the two investigators (P.L. and C.T.). Inconsistencies were discussed with the third reviewer (W.C.) and solved by common agreement. The size of the study did not limit inclusion.

### 2.3. Data Collection Process

A structured data collecting form was used to collect the following data from individual studies including title, name of authors, year of the study, publication year, country where the study was conducted, patient characteristics, AKI definition, incidence of AKI, incidence of severe AKI requiring RRT, and mortality risk of AKI among patients after lung transplantation.

### 2.4. Statistical Analysis

Comprehensive Meta-Analysis software version 3.3.070 (Biostat Inc., Englewood, NJ, USA) was used to perform meta-analysis. Adjusted point estimates of included studies were incorporated by the generic inverse variance method of DerSimonian–Laird, which assigned the weight of an individual study based on its variance [34]. Due to the probability of between-study variance, we applied a random-effects model to pool outcomes of interest, including incidence of AKI and mortality risk. Statistical heterogeneity of studies was assessed by the Cochran’s Q test (*p* < 0.05 for a statistical significance) and the I^2^ statistic (≤25% represents insignificant heterogeneity, 26% to 50% represents low heterogeneity, 51% to 75% represents moderate heterogeneity, and ≥75% represents high heterogeneity) [35]. The presence of publication bias was assessed by both funnel ploy and the Egger test [36].

## 3. Results

The search yielded a total of 1809 articles for initial screening. After removal of 714 duplicates, the titles and abstracts of 1095 articles were screened for eligibility. A total of 922 articles were excluded (due to in vitro studies, pediatric patient population, animal studies, case reports, correspondences, or review articles). A total of 173 potentially relevant studies were included for full-length article review; 147 of them were additionally excluded from the full-text review as they did not provide the outcome of interest (*n* = 77) or were not observational studies (*n* = 48), or did not use a standard AKI definition (*n* = 22).

Thus, 26 cohort studies [10,11,13,14,19,21,24,28,29,37,38,39,40,41,42,43,44,45,46,47,48,49,50,51,52,53] with 40,592 patients undergoing lung transplantation were identified. Figure 1 shows a flowchart outlining identification of papers for inclusion. Table 1 presents the characteristics of the included studies. The kappa for systematic searches, selection of studies and data extraction were 0.98, 0.87, and 0.98, respectively.

### 3.1. Incidence of Acute Kidney Injury among Patients after Lung Transplantation

The pooled estimated incidence rates of AKI and severe AKI requiring RRT after lung transplantation were 52.5% (95% CI: 45.8–59.1%, I^2^ = 89%, Figure 2) and 9.3% (95% CI: 7.6–11.4%, I^2^ = 90%, Figure 3). Subgroup analysis based on the AKI definition was performed and demonstrated a pooled estimated incidence of AKI after lung transplantation of 49% (95% CI: 38.3–59.8%, I^2^ = 86%, Figure 2) by RIFLE criteria, 55.5% (95% CI: 45.2–65.4%, I^2^ = 71%, Figure 2) by AKIN criteria, and 53.0% (95% CI: 38.2–67.3%, I^2^ = 91%, Figure 2) by KDIGO criteria.

Meta-regression analysis demonstrated that year of study did not significantly affect the incidence of AKI (*p* = 0.22) and severe AKI requiring RRT (*p* = 0.68) among patients after lung transplantation, as shown in Figure 4.

### 3.2. Mortality Risk of Acute Kidney Injury in Patients after Lung Transplantation

Data on mortality risk from included studies are shown in Table 2. The pooled OR of hospital mortality among patients after lung transplantation with AKI and severe AKI requiring RRT were 2.75 (95% CI, 1.18–6.41, I^2^ = 69%, Figure 5A) and 10.89 (95% CI, 5.03–23.58, I^2^ = 82%, Figure 5B), respectively. At one year, the pooled OR of mortality among patients after lung transplantation with AKI and severe AKI requiring RRT were 2.99 (95% CI, 1.72–5.18, I^2^ = 74%, Appendix A) and 8.32 (95% CI, 5.95–11.63, I^2^ = 70%, Appendix A), respectively. At five years, the pooled OR of mortality among patients after lung transplantation with AKI and severe AKI requiring RRT were 1.47 (95% CI, 1.11–1.94, I^2^ = 0%, Appendix A) and 4.79 (95% CI, 3.58–6.40, I^2^ = 81%, Appendix A), respectively.

### 3.3. Evaluation for Publication Bias

Funnel plot (Appendix A) and Egger’s regression asymmetry tests were performed to assess publication bias in analysis evaluating mortality risk of AKI in patients after lung transplant with AKI and severe AKI requiring RRT, respectively. We found no significant publication bias in meta-analysis evaluating mortality risk of patients after lung transplant with AKI (*p* = 0.99) and severe AKI requiring RRT (*p* = 0.50).

## 4. Discussion and Conclusions

In this systematic review, we demonstrated that AKI in patients after lung transplantation is common, with pooled incidence rates of AKI and severe AKI requiring RRT in patients after lung transplantation of 52.5% and 9.3%, respectively. We also showed that the incidence of AKI in patients after lung transplantation has not improved, despite advances in therapy. Compared to those without AKI, patients with post-lung-transplant AKI had increased short and long-term mortality.

Some specific factors related to AKI after lung transplant include hypercapnia/hypoxemia-mediated impaired renal blood flow (RBF), hemodynamics during lung transplant surgery, and the use of extracorporeal membrane oxygenation (ECMO) and cardio-pulmonary bypass (CPB) during lung transplant surgery [12,28,44,54]. Postoperative respiratory failure is common after lung transplantation; reported to be as high as 55% [55]. Hypoxemia is associated with reduced RBF in a dose-dependent relationship [56,57,58] thought to be related to stimulation of adrenergic neurons and alterations in nitric oxide metabolism [59]. In addition, studies have shown that hypercapnia can induce peripheral vasodilatation and decreased systemic vascular resistance, with a compensatory neurohormonal vasoconstriction response. This leads to activation of the renin-angiotensin-aldosterone system (RAAS) and direct renal vasoconstriction, resulting in a reduction in RBF and GFR [56,60,61,62]. Furthermore, poorly controlled hemodynamics during lung transplant surgery can result in intraoperative hypotension, one of the most significant risk factors for the development of AKI after lung transplantation [10,24]. Currently, CPB remains a standard method used in lung transplantation for intraoperative cardiorespiratory support, especially in cases of poor hemodynamic tolerance or severe pulmonary arterial hypertension [63]. However, CPB is commonly associated with inflammatory reactions and bleeding complications [64]. ECMO has more recently been used as an alternative option to CPB for intraoperative cardiopulmonary support during lung transplantation [63]. When compared to CPB, studies have demonstrated beneficial effects of intraoperative ECMO support, with lower rates of primary graft dysfunction, acute post-operative bleeding, AKI requiring RRT, and length of hospital stay [63]. However, the use of ECMO itself may also cause a renal insult related to the activation of proinflammatory mediators caused by the continuous exposure of blood to the non-biological and non-endothelialized ECMO interface [65]. Therefore, our study demonstrated that patients undergoing lung transplantation more frequently develop AKI and AKI requiring RRT than abdominal solid organ transplantation, such as liver transplantation (incidence of AKI and AKI requiring RRT of 41% and 7%, respectively) [66].

As there are currently no effective targeted pharmacotherapies available for AKI, treatment is limited to supportive strategies and RRT when indicated [4,5,6,8]. Patients who recover from AKI continue to have an increased risk of mortality on either short or long-term follow-up [9]. Following post-lung-transplant AKI, patients may develop CKD, with rates of progression to ESKD as high as 3.8%, 7.2%, and 7.9% at one, five, and ten years after lung transplant, respectively [17,46,67]. Therefore, prevention and early identification of AKI in patients at risk for post-lung-transplant AKI may potentially play an important role in improving patient outcomes. Important risk factors for AKI in patients after lung transplantation include bilateral lung transplantation [19,21,29,68], lower baseline estimated GFR [13,19,38,46,68], pulmonary hypertension [19,38,46], duration of mechanical ventilation requirement [13,14,24,28,46,53], the need for ECMO support [19,46,68], intraoperative hypotension, and vasopressor requirement [10,24] (Table 3).

There is experimental data that injurious ventilation strategies, such as a high tidal volume, low positive end-expiratory pressure approach can cause renal epithelial cell apoptosis and dysregulation of extracellular ligands that control renal vascular tone and endothelial integrity, resulting in AKI [69,70]. Among patients with acute respiratory distress syndrome, protective lung ventilation is associated with a reduced risk for AKI requiring dialysis and improves dialysis-free survival [71]. Therefore, future studies are needed to assess whether maintaining perioperative lung-protective ventilation helps to prevent AKI following lung transplantation [12]. Moreover, ECMO management may also play an important role in the prevention of post lung transplant AKI. High ECMO pump speed is associated with hemolysis and AKI development [65]. Future prospective studies are needed to assess the effects of ECMO pump speed on the risk of post-lung-transplantation AKI. Finally, immunosuppressive medications may also play an important role in AKI development following lung transplantation [12,72]. AKI related to calcineurin inhibitor (CNI)-induced thrombotic microangiopathy (TMA) has been reported in lung transplant recipients and is often missed or recognized late in the ICU setting [12,72]. Although TMA in lung transplant recipients is a rare condition, early recognition and management can potentially reduce post lung transplant-related morbidity and mortality [12,72,73].

Our study has some limitations. Firstly, there are statistical heterogeneities in our meta-analysis. Subgroup analyses were performed using differing AKI definitions, including RIFLE criteria AKIN criteria and KDIGO criteria. Meta-regression analysis assessing the effect of year of study on the incidence of AKI was also performed, and we found no significant correlation between year of study and incidence of AKI post lung transplantation. Secondly, AKI diagnosis from the included studies was based on changes in serum creatinine, and data on urine output and AKI biomarkers was limited. Lastly, this systematic review is primarily based on observational studies, as the data from clinical trials or population-based studies were limited. Thus, it can, at best, demonstrate an association between AKI and increased short-term and long-term mortality post lung transplant, but not a causal relationship.

In summary, AKI and severe AKI requiring RRT are common in patients after lung transplantation, with overall estimated incidence rates of 52.5% and 9.3%, respectively. Post-lung-transplant AKI is significantly associated with reduced short-term and long-term survival. Despite advances in transplantation therapy, the incidence of AKI in patients after lung transplantation does not appear to have improved.

## Figures and Tables

**Figure 1 jcm-08-01713-f001:**
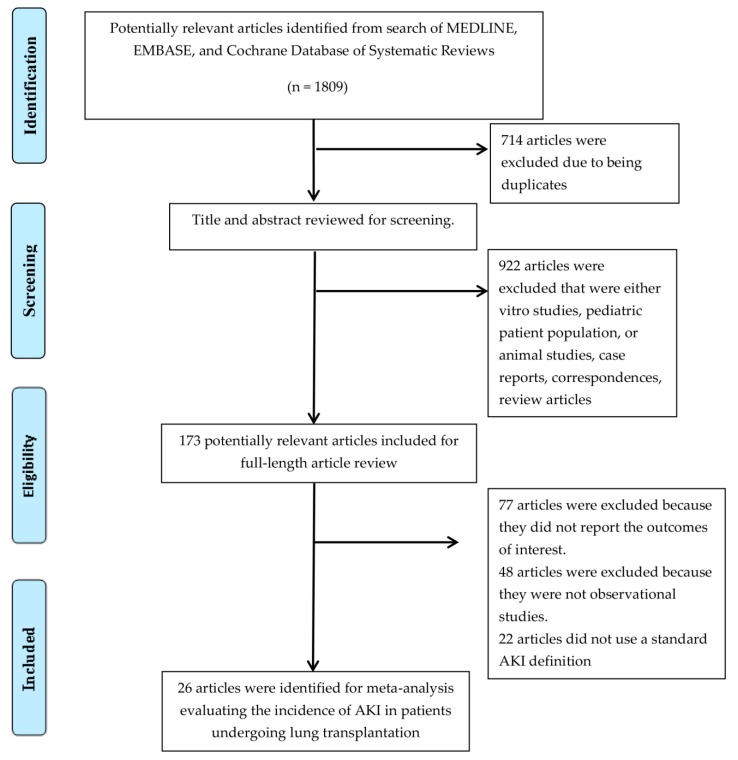
Outline of our search methodology. AKI, Acute kidney injury.

**Figure 2 jcm-08-01713-f002:**
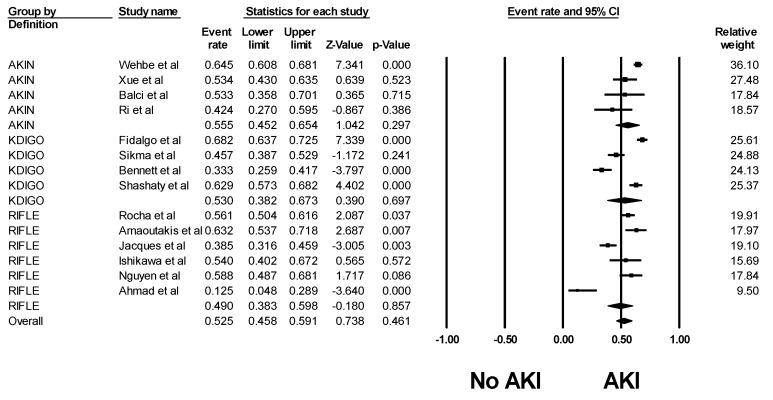
Forest plots of the included studies evaluating incidence rates of AKI after lung transplantation. AKI, Acute kidney injury.

**Figure 3 jcm-08-01713-f003:**
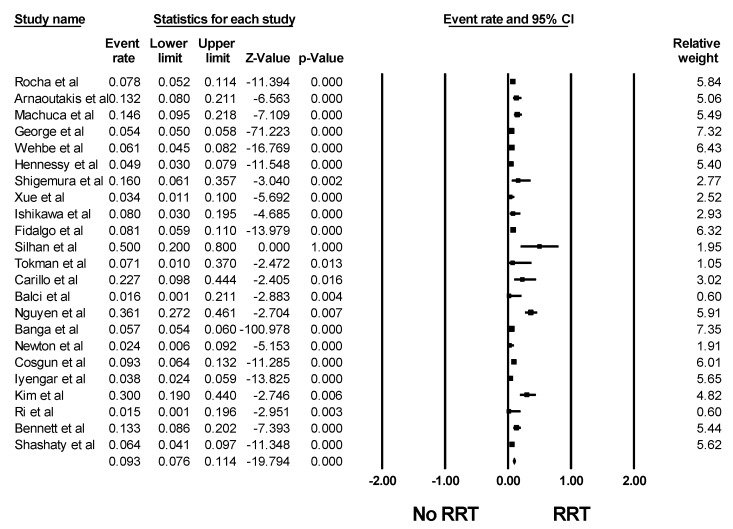
Forest plots of the included studies evaluating incidence rates of AKI requiring RRT after lung transplantation. AKI, Acute kidney injury; RRT, renal replacement therapy.

**Figure 4 jcm-08-01713-f004:**
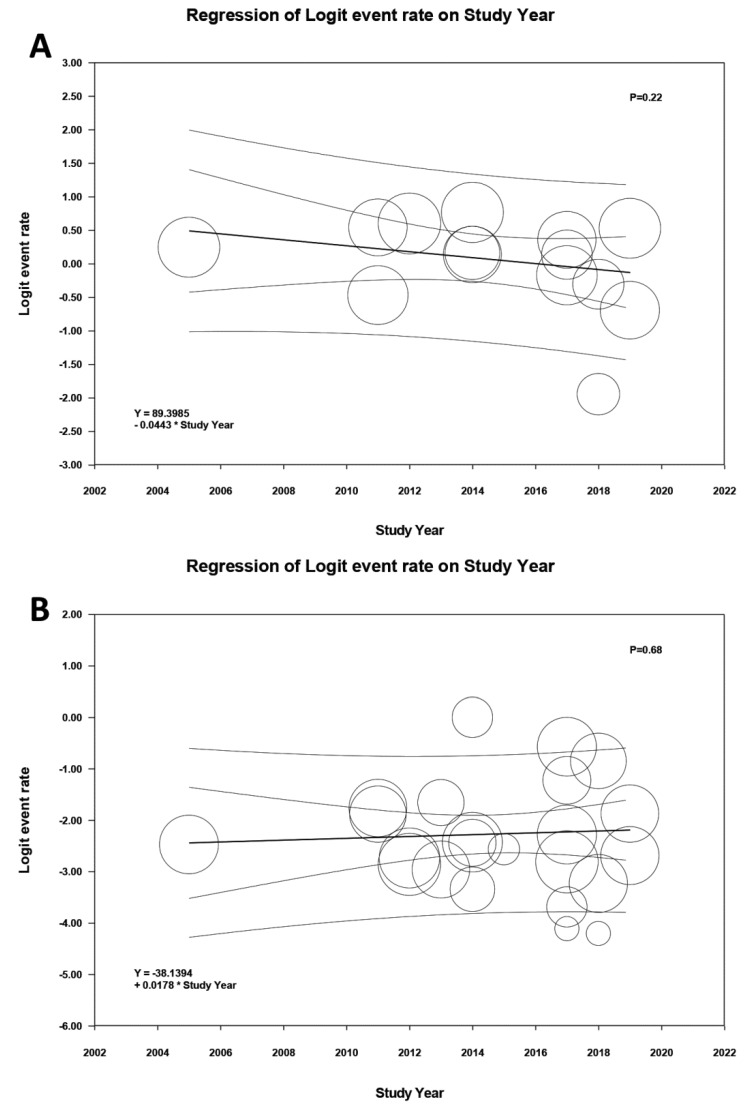
Meta-regression analyses showed that year of study did not significantly affect (**A**) the incidence of AKI (*p* = 0.11) and (**B**) severe AKI requiring RRT (*p* = 0.54) among patients after lung transplantation. AKI, acute kidney injury; RRT, renal replacement therapy.

**Figure 5 jcm-08-01713-f005:**
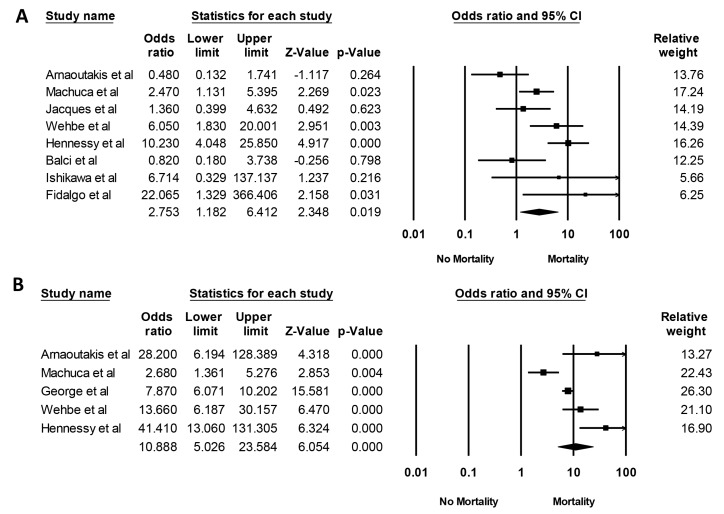
Forest plots of the included studies evaluating hospital mortality of (**A**) AKI and (**B**) AKI requiring RRT after lung transplantation. AKI, acute kidney injury; RRT, renal replacement therapy.

**Table 1 jcm-08-01713-t001:** Main characteristic of studies included in analysis assessing the incidence of acute kidney injury in patients after lung transplantation.

Study	Year	Country	Patients	Number	AKI Definition	AKI Incidence
Rocha et al. [13]	2005	USA	Patents underwent lung transplantation	296	RIFLE criteria	AKI 166/296 (56.1%) RRT 23/296 (7.8%)
			-Double 146/296 (49.3%)			
			-COPD 134/296 (45.3%)			
			-Cystic fibrosis 61/296 (20.6%)			
			-Idiopathic pulmonary fibrosis 31/296 (10.5%)			
Arnaoutakis et al. [11]	2011	USA	Patients underwent lung transplantation	106	RIFLE criteria	AKI 67/106 (63.2%) RRT 14/106 (13.2%)
			-Double 93/106 (87.7%)			
			-COPD 33/106 (31.1%)			
			-Idiopathic pulmonary fibrosis 22/106 (20.8%)			
			-Cystic fibrosis 21/106 (19.8%)			
Machuca et al. [37]	2011	Brazil	Patients underwent lung transplantation	130	Doubling of baseline serum creatinine levels	AKI 41/130 (31.5%) RRT 19/130 (14.6%)
			- Idiopathic pulmonary fibrosis 53/130 (40.8%)			
			-COPD 52/130 (40%)			
			-Lymphangioleiomyomatosis 8/130 (6.2%)			
			-Cystic fibrosis 4/130 (3%)			
George et al. [19]	2012	USA	Patients underwent lung transplantation from UNOS database	12,108	RRT	RRT 655/12,108 (5.41%)
			-Double 6876/12,108 (56.8%)			
			-COPD 4227/12,108 (34.9%)			
			-Idiopathic pulmonary fibrosis 3369/12,108 (27.8%)			
Jacques et al. [21]	2012	Canada	Patients underwent lung transplantation	174	RIFLE criteria	AKI 67/174 (38.5%)
			-Double 85/174 (58.9%)			
			-Emphysema 64/174 (36.8%)			
			-Cystic fibrosis 44 /174 (25.3%)			
			-Idiopathic pulmonary fibrosis 24/174 (13.8%)			
Wehbe et al. [38]	2012	USA	Patients underwent lung transplantation	657	AKIN classification	AKI 424/657 (64.5%) RRT 40/657 (6.1%)
			-Double 372/657 (56.6%)			
			-COPD 233/657 (35.5%)			
			-Idiopathic pulmonary fibrosis 212/657 (32.3%)			
			-Cystic fibrosis 90/657 (13.7%)			
Hennessy et al. [39]	2013	USA	Patients underwent lung transplantation	352	SCr > 3 mg/dL within 5 days after surgery	AKI 33/325 (9.4%) RRT 16/325 (4.9%)
			-Double 98/352 (27.8%)			
			-COPD 170/352 (48.2%)			
			-Cystic fibrosis28/352 (8%)			
			-Pulmonary fibrosis 53/352 (15%)			
Shigemura et al. [40]	2013	USA	-Patients underwent lobar lung transplantation.	25	RRT	RRT 4/25 (16%)
			-Double 13/25 (52%)			
			-Idiopathic pulmonary fibrosis 9/25 (36%)			
			-Sarcoidosis 4/25 (16%)			
			-Cystic fibrosis 2/25 (8%)			
Xue et al. [28]	2014	China	Patients underwent lung transplantation	88	AKIN classification	AKI 47/88 (53.4%) RRT 3/88 (3.4%)
			-Double 38/88 (43.2%)			
			-Idiopathic pulmonary fibrosis 46/88 (52.3%)			
			-COPD 19/88 (21.6%)			
			-Brochiectesis7/88 (8%)			
Ishikawa et al. [29]	2014	Canada	Patients underwent lung transplantation	50	RIFLE criteria	AKI during first 72 hours after transplant 27/50 (54%) AKI during hospitalization 32/50 (64%) RRT 4/50 (8%)
			-Double 15/50 (30%)			
			-Interstitial lung disease 18/50 (36%)			
			-COPD 14/50 (28%)			
			-Cystic fibrosis 10/50 (20%)			
			-Alpha 1 antitrypsin deficiency 5/50 (10%)			
Fidalgo et al. [14]	2014	Canada	Patient underwent lung transplant	445	KDIGO criteria	Total AKI 306/445 (68.8%) AKI in lung transplant only 290/425 (68.2%) RRT 36/445 (8.1%)
			-Double 354/445 (79.6%)			
			-Heart-lung transplant 20/445 (4.5%)			
			-COPD 149/445 (33.5%)			
			-Idiopathic pulmonary fibrosis 99/445 (22.2%)			
			-Cystic fibrosis 71/445 (16%)			
Silhan et al. [41]	2014	USA	Pulmonary fibrosis patients with telomerase mutation carriers underwent lung transplant	8	RRT	RRT 4/8 (50%)
			-Double 5/8 (62.5%)			
Tokman et al. [42]	2015	USA	Pulmonary fibrosis patients underwent lung transplantation	14	Increase in serum creatinine of ≥ 1.5 times from baseline within seven days after transplant	AKI 8/14 (57.1%) RRT 1/14 (7.1%)
			-Double 12/14 (85.7%)			
Sikma et al. [44]	2017	Netherlands	Patient underwent lung transplantation	186	KDIGO criteria	AKI 85/186 (45.7%)
			-Double 148/186 (79.6%)			
			-COPD/alpha 1 antitrypsin deficiency 80/186 (43%)			
			-Sarcoidosis/Interstitial lung disease/usual interstitial pneumonia 14/186 (7.5%)			
Carillo et al. [43]	2017	Italy	Patients underwent lung transplantation	22	RRT	RRT 5/22 (22.7%)
			-Double 6/22 (27.3%)			
			-Pulmonary fibrosis 13/22 (59%)			
			-Emphysema 7/22(31.8%)			
Balci et al. [24]	2017	Turkey	Patients underwent lung transplantation	30	AKIN classification	AKI 16/30 (53.3%) RRT 0/30 (0%)
			-Idiopathic pulmonary fibrosis 10/30 (33.3%)			
			-COPD 6/30 (20%)			
			-Cystic fibrosis/bronchiectasis 9/30 (30%)			
Nguyen et al. [10]	2017	USA	Patients underwent lung transplantation	97	RIFLE criteria	AKI 57/97 (58.8%) RRT 35/97 (38.5%)
			-Double 55/97 (56.7%)			
			-COPD 11/97 (11.3%)			
			-Idiopathic pulmonary fibrosis 50/97 (51.5%)			
			-Cystic fibrosis 20/97 (20.6%)			
			-Pulmonary hypertension 11/97 (11.3%)			
Banga et al. [46]	2017	USA	Patients underwent lung transplantation. Data was from UNOS database from 1994–2014	24,110	RRT	RRT 1369/24,110 (5.7%)
Newton et al. [45]	2017	USA	Pulmonary fibrosis patients underwent lung transplantation.	82	increase in serum creatinine to ≥ 1.5 times from baseline within 7 days after transplant	AKI 54/82 (65.9%) RRT 2/82 (2.4%)
			-Double 70/82 (85.4%)			
Cosgun et al. [47]	2017	USA	Patient underwent lung transplantation	291	RRT	RRT 27/291 (9.3%)
			-Double 285/291 (97.9%)			
			-Requiring intraoperative ECMO 134/291 (46.0%)			
			-Cystic fibrosis 89/291 (30.6%)			
			-COPD 88/291 (30.2%)			
			-Idiopathic pulmonary fibrosis 63/291 (21.6%)			
Ahmad et al. [48]	2018	USA	Patients underwent lung transplantation from brain death donors	32	RIFLE criteria	AKI at 24 hours post-transplant = 6/32 (18.8%) AKI at 72 hours post-transplant = 4/32 (12.5%)
			-Double 20/32 (62.5%)			
			-Interstitial lung disease 24/32 (75%)			
			-COPD 8/32 (25%)			
Iyengar et al. [49]	2018	USA	Patients underwent lung transplantation	501	RRT	RRT 19/501 (3.8%)
			-Double 267/501 (53.3%)			
Ri et al. [50]	2018	Korea	Patient underwent lung transplantation	33	AKIN criteria	AKI 14/33 (42.4%) RRT 0/33 (0%)
			-Idiopathic pulmonary fibrosis 12/33 (36.4%)			
			-Interstitial lung disease 20/33 (60.6%)			
			-Primary pulmonary hypertension 1/14 (7.1%)			
Calabrese et al. [51]	2018	USA	Patient underwent lung transplantation	321	KDIGO criteria	AKI KDIGO stage 2 and 3 61/321 (19.0%)
			-Double 288/321 (89.7%)			
			-Heart-lung 6/321 (1.9%)			
			-Idiopathic pulmonary fibrosis 210/321 (65.4%)			
			-COPD 66/321 (20.6%)			
			-Cystic fibrosis 31/321 (9.7%)			
Bennett et al. [52]	2019	Italy	Patients underwent lung transplantation	135	KDIGO criteria	AKI 45/135 (33.3%) RRT 18/135 (13.3%)
			-Double 66/135 (48.9%)			
			-Pulmonary fibrosis 72/135 (53.33%)			
			-COPD 28/135 (20.74%)			
			-Cystic fibrosis 25/135 (18.52%)			
Shashaty et al. [53]	2019	USA	Patients underwent lung transplantation	299	KDIGO criteria	AKI 188/299 (62.9%) RRT 19/299 (6.4%)
			-Double 180/299 (60.2%)			
			-COPD 119/299 (39.8%)			
			-Interstitial lung disease 123/299 (41.1%)			
			-Cystic fibrosis 26/299 (8.70%)			

Abbreviations: AKI, Acute kidney injury; AKIN, acute kidney injury network; COPD, chronic obstructive pulmonary disease; KDIGO, kidney disease improving global outcomes; RIFLE, risk, injury, failure, loss of kidney function, and end-stage kidney disease; RRT, renal replacement therapy; UNOS, United Network for Organ Sharing; USA, United States of America; SCr, serum creatinine.

**Table 2 jcm-08-01713-t002:** Mortality risk of AKI in patients after lung transplantation.

Study	Year	Results	Confounder Adjustment
Rocha et al. [13]	2005	One-year mortality	None
		AKI: 4.33 (2.08–8.99)	
		RRT: 23.70 (8.29–67.80)	
		Five-year mortality	
		AKI: 1.44 (0.90–2.30)	
		RRT: 9.73 (2.82–33.53)	
Arnaoutakis et al. [11]	2011	In-hospital mortality	Lung allocation score, pre-transplant GFR, recipient age, donor cigarette use, postoperative tracheostomy
		AKI: 0.48 (0.13–1.71)	
		RRT: 28.2 (6.18–128.1)	
		One-year mortality	
		AKI: 0.47 (0.20–1.14)	
		RRT: 4.97 (1.54–16.0)	
Machuca et al. [37]	2011	Mortality	Mechanical ventilation duration, reintubation, acute rejection in the first month, coronary heart disease
		AKI: 2.47 (1.13–5.39)	
		RRT: 2.68 (1.36–5.27)	
George et al. [19]	2012	30-day mortality	Recipient age, GFR, BMI, diagnosis, mean pulmonary artery pressure, ICU status, ECMO support, donor age, bilateral lung transplant, ischemic time, annual center volume
		RRT: 7.87 (6.07–10.20)	
		One-year mortality	
		RRT: 7.89 (6.80–9.15)	
		Five-year mortality	
		RRT: 5.35 (4.72–6.07)	
Jacques et al. [21]	2012	30-day mortality	Age, sex, indication, ICU length of stay, coronary artery disease, aprotinin use, double lung transplantation
		AKI:1.36 (0.40–4.64)	
		Long-term mortality	
		AKI: 1.54 (0.79–2.99)	
Wehbe et al. [38]	2012	Hospital mortality	Age, sex, race, type on lung transplantation, COPD, pre-transplantation diabetes, baseline creatinine
		AKI: 6.05 (1.83–20.00)	
		RRT: 13.66 (6.19–30.17)	
		One-year mortality	
		AKI: 2.92 (1.76–4.82)	
Hennessy et al. [39]	2013	30-day mortality	None
		AKI: 10.23 (4.05–25.86)	
		RRT: 41.41 (13.06–131.31)	
		One-year mortality	
		AKI: 7.01 (3.29–14.92)	
		RRT: 43.04 (9.48–195.50)	
Xue et al. [28]	2014	Five-year mortality	Age, sex, type, and cause of lung transplant, hypertension, and diabetes
		AKI: 1.48 (1.04–2.11)	
Ishikawa et al. [29]	2014	Mortality	None
		AKI: 3/27 (11%) vs. 0/23 (0%)	
Fidalgo et al. [14]	2014	Hospital mortality	Age, sex, COPD, eGFR, LAS score, diabetes mellitus, pulmonary artery pressure, previous sternotomy, type of lung transplant, ICU length of stay
		AKI: 22/306 (7%) vs. 0/139 (0%)	
		One-year mortality	
		AKI: 2.81 (1.15–6.84)	
Balci et al. [24]	2017	30-day mortality	None
		AKI: 0.82 (0.18–3.74)	
Nguyen et al. [10]	2017	One-year mortality	None
		AKI: 1.73 (0.42–7.13)	
		RRT: 1.20 (0.32–4.60)	
Banga et al. [46]	2017	One-year mortality	Age, serum albumin, type of procedure, CMV mismatch, Length of hospital stay after transplantation, recipient hospitalized at the time of transplant, history of prior cardiac surgery, acute rejection
		RRT: 7.23 (6.2–8.43)	
		Five-year mortality	
		RRT: 3.96 (3.43–4.56)	
Bennett et al. [52]	2019	One-year mortality	None
		AKI: 6.20 (2.74–14.05)	
		RRT: 21.60 (5.75–81.11)	
Shashaty et al. [53]	2019	One-year mortality	Primary graft dysfunction, age, bilateral lung transplant
		AKI: 3.64 (1.68–7.88)	

Abbreviations: AKI, acute kidney injury; RRT, renal replacement therapy; BMI, body mass index; CMV, cytomegalovirus; COPD, chronic obstructive pulmonary disease; DM, diabetes mellitus; ECMO, extracorporeal membrane oxygenation; eGFR, estimated glomerular filtration rate; HES, hydroxyethyl starch; ICU, intensive care unit; SCr, serum creatinine.

**Table 3 jcm-08-01713-t003:** Reported risk factors for AKI after lung transplantation.

Reported Risk Factors for AKI after Lung Transplantation
High baseline SCr, lower baseline eGFR [13,19,38,46,68] Male [52] Older age [19] Higher BMI [68] Carriers of the ABCB1 CGC-CGC diplotype [51] African American and Hispanic ethnicity [19,46,68] Higher mean pulmonary artery pressure, pulmonary hypertension [19,38,46] Intraoperative hypoxemia [29] Duration of mechanical ventilation requirement [13,14,24,28,46] Duration of ICU stay [68] The need of ECMO support [19,46,68] Bilateral lung transplantation [19,21,29,68] Non-COPD diagnosis [13] Cystic fibrosis [44,53] Idiopathic pulmonary fibrosis [19] Sarcoidosis [68] Intraoperative hypotension and vasopressors requirement [10,24] Higher HES volume [29] Longer cardiopulmonary bypass time [10,14,52] Ischemic time [19] Aprotinin use [21] Pretransplant diabetes mellitus [38,68] Pretransplant hypertension [28] Longer time on waiting list [68] Lower Karnofsky performance score [68] Supratherapeutic cyclosporine/tacrolimus trough [14,44] Amphotericin B use [13] and nephrotoxic medications [44] Infection [24,44]

Abbreviations: BMI, body mass index; COPD, chronic obstructive pulmonary disease; DM, diabetes mellitus; ECMO, extracorporeal membrane oxygenation; eGFR, estimated glomerular filtration rate; HES, hydroxyethyl starch; ICU, intensive care unit; SCr, serum creatinine.

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
