# Peer review of "Acute Kidney Injury after Lung Transplantation: A Systematic Review and Meta-Analysis"

_jcm, 2019, doi:10.3390/jcm8101713_

Round 1

Reviewer 1 Report

Dear Authors,

The article entitled “Acute kidney injury after Lung Transplantation: A Systematic Review and Meta-analysis” (Manuscript ID: jcm-600293) is very well quantity review. Authors put in a huge effort to analyze literature data. The search yielded a total of 1,809 articles for initial screening, but finally was analyzed 26 cohort studies with 40,592 patients undergoing lung transplantation (many articles were excluded due to various factors discussed in the text and shown in figure 1). It should be noted that the analysis was carried out independently by two researchers, any Inconsistencies were discussed with the third investigator.

Methods (Information Sources and Search Strategy; Study Selection; Data Collection Process; Statistical analysis) have been fairly and clearly described.

The results of the meta-analysis were very well discussed in the text and clearly presented in the 11 Figures (the main – 5 and supplementary - 6) in the 3 Tables.

The obtained results were aptly and matter-of-factly discussed in the Discussion part based on well-chosen literature.

In my opinion, this meta-analysis gives valuable and unfortunately, quite pessimistic information that incidence rates of  acute kidney injury (AKI) and severe AKI requiring renal replacement therapy (RRT) in patients after lung transplantation are 52.5% and 9.3%, respectively and there is not advances therapy that could protect patients after lung transplantation from AKI and severe AKI requiring RRT. Also Authors have shown that AKI after lung transplantation is significantly associated with reduced short-term and long-term survival.

Despite the fact that the work is very valuable, I have some comments to the authors:

In Abstract: Background: there should be also mentioned about AKI requiring renal replacement therapy, so that the abbreviation AKI requiring RRT would be explained in the abstract. In Abstract: Methods: there should be also add (before the word MEDLINE) Ovid. Eeditorial corrections: In Results (on page3), in second paragraph – the last statement should be ended by dot (dot lack) In description of Figure 1. there is too much dots ate the end of statement. The table showing in Figure 2. In column “group by definition”, in last row contains word “Overall”- is good, that’s why other tables (Figure 2, Figure 5AB, Supplementary Figure 1, Supplementary Figure 2, Supplementary Figure 3 and Supplementary Figure 4) have not Overall? Description of Supplementary Figures in the manuscript body should be not bolded. In Table 3., the column Results should be wider, then the data contained therein would look nicer. In 1. Incidence of Acute Kidney Injury among Patients after Lung Transplantation, …..more space before brackets

55.5% (95%CI: …..

53.0% (95%CI:………

Author Response

Response to Reviewer

The article entitled “Acute kidney injury after Lung Transplantation: A Systematic Review and Meta-analysis” (Manuscript ID: jcm-600293) is very well quantity review. Authors put in a huge effort to analyze literature data. The search yielded a total of 1,809 articles for initial screening, but finally was analyzed 26 cohort studies with 40,592 patients undergoing lung transplantation (many articles were excluded due to various factors discussed in the text and shown in figure 1). It should be noted that the analysis was carried out independently by two researchers, any Inconsistencies were discussed with the third investigator.

Methods (Information Sources and Search Strategy; Study Selection; Data Collection Process; Statistical analysis) have been fairly and clearly described.

The results of the meta-analysis were very well discussed in the text and clearly presented in the 11 Figures (the main – 5 and supplementary - 6) in the 3 Tables.

The obtained results were aptly and matter-of-factly discussed in the Discussion part based on well-chosen literature.

In my opinion, this meta-analysis gives valuable and unfortunately, quite pessimistic information that incidence rates of acute kidney injury (AKI) and severe AKI requiring renal replacement therapy (RRT) in patients after lung transplantation are 52.5% and 9.3%, respectively and there is not advances therapy that could protect patients after lung transplantation from AKI and severe AKI requiring RRT. Also Authors have shown that AKI after lung transplantation is significantly associated with reduced short-term and long-term survival.

Despite the fact that the work is very valuable, I have some comments to the authors:

Response: We thank you for reviewing our manuscript and for your critical evaluation. We really appreciated your input and found your suggestions very helpful.

Comment #1

In Abstract: Background: there should be also mentioned about AKI requiring renal replacement therapy, so that the abbreviation AKI requiring RRT would be explained in the abstract.

Response: We agree with the reviewer. Acute kidney injury requiring renal replacement therapy was mentioned in background section of abstract as suggested.

Comment #2

In Abstract: Methods: there should be also add (before the word MEDLINE) Ovid.

Response: We appreciate the reviewer’s input. We agree and MEDLINE has been revised to Ovid MEDLINE as suggested

Comment #3

In Results (on page3), in second paragraph – the last statement should be ended by dot (dot lack)

Response: We agree with the reviewer. Full stop has been added at the end of the sentence as suggested.

Comment #4

In description of Figure 1. there is too much dots ate the end of statement.

Response: We agree with the reviewer. Extra full stop has been deleted as suggested.

Comment #5

The table showing in Figure 2. In column “group by definition”, in last row contains word “Overall”- is good, that’s why other tables (Figure 2, Figure 5AB, Supplementary Figure 1, Supplementary Figure 2, Supplementary Figure 3 and Supplementary Figure 4) have not Overall?

Response: The reviewer raised very important point. Unfortunately in “Overall” is shown only in the Figure 2 because there is a subgroup analysis. It is a limitation in the program Comprehensive meta-analysis and thus we are not able to add overall in other figures due the limitation of the program.

Comment #6

Description of Supplementary Figures in the manuscript body should be not bolded.

Response: We agree with the reviewer. This has been corrected as suggested.

Comment #7

In Table 3, the column Results should be wider, then the data contained therein would look nicer.

Response: The width of Results column of Table 3 has been increased as suggested.

Comment #8

In 1. Incidence of Acute Kidney Injury among Patients after Lung Transplantation, …..more space before brackets

Response: We agree with the reviewer. Space has been added before brackets as suggested.

We greatly appreciated the reviewer’s time and comments to improve our manuscript.

Reviewer 2 Report

The authors conducted an interesting review and meta-analysis, the methodology seems to be adequate and de development of the meta-analysis correct. Despite this, I would like to clarify the following aspects:

INTRODUCTION:

The definition of AKI should be clear and concise, if not possible due to the different definitions available, the authors should cite at least all the definitions accepted in their study.

All data supporting the present study should be clear in the introduction, the sentence "Many lung transplant..." (line 62) can not be accepted, the authors should be clear about the epidemiology of this disease.

Line 74, again, the expression "standard AKI definitions" must be supported by citations.

RESULTS:

The authors show the reported risk factors for AKI after lung transplantation but apparently these data are not included in the analysis. Can the authors provide any information about the influence of these risk factors in the analysed studies? Were these variables included in the pooled analysis? It would be of great interest if the authors could explain and compare the potential influence of each one of these factors in the development of AKI after lung transplantation or at least provide the frequency of each on eof the described factors in the analysed studies.

If the authors are not able to provide more information regarding these variables the Table 2 should be removed or changed to the Introduction section since it can not be considered as a "Result" of their work.

DISCUSSION:

This section should be deeply revised by the authors. The first paragraph of the discussion could be removed, since it talks about pathophysiology of the disease, this is not a discussion.

There are many data in the discussion section that were provided in table 2 and should be not be duplicated.

Authors should analyse their results, compare the included studies and try to explain, for example, the huge variability between the analysed studies regarding the development of AKI, risk factors, possible causes for these differences...

Author Response

Response to Reviewer

The authors conducted an interesting review and meta-analysis, the methodology seems to be adequate and de development of the meta-analysis correct. Despite this, I would like to clarify the following aspects:

Response: We thank you for reviewing our manuscript and for your critical evaluation. We really appreciated your input and found your suggestions very helpful.

Comment #1

INTRODUCTION:

The definition of AKI should be clear and concise, if not possible due to the different definitions available, the authors should cite at least all the definitions accepted in their study

Response: We agree with the reviewer. RIFLE, AKIN, and KDIGO definition of AKI was stated and properly cited in the introduction as the reviewer’s suggestion.

Comment #2

All data supporting the present study should be clear in the introduction, the sentence "Many lung transplant..." (line 62) cannot be accepted, the authors should be clear about the epidemiology of this disease.

Response: We appreciate the reviewer’s input. We agree with the reviewer. The statement has been revised as suggested.

Up to 68% of lung transplant recipients develop AKI, which has been associated with increased 1-year mortality, length of hospital stay, higher resource utilization, and related health care burden [10-22].

Comment #3

Line 74, again, the expression "standard AKI definitions" must be supported by citations.

Response: We agree with the reviewer. RIFLE, AKIN, and KDIGO definition of AKI was stated and properly cited as suggested

Comment #4

RESULT

The authors show the reported risk factors for AKI after lung transplantation but apparently these data are not included in the analysis. Can the authors provide any information about the influence of these risk factors in the analysed studies? Were these variables included in the pooled analysis? It would be of great interest if the authors could explain and compare the potential influence of each one of these factors in the development of AKI after lung transplantation or at least provide the frequency of each on of the described factors in the analysed studies.

If the authors are not able to provide more information regarding these variables the Table 2 should be removed or changed to the Introduction section since it cannot be considered as a "Result" of their work.

Response: We agree with the reviewer. The reviewer raised very important point. We apologize, due to limited data, we cannot perform analysis as suggested. We agree with the reviewer and the Table of risk factors has now been moved to discussion as suggested.

Comment #5

DISCUSSION:

This section should be deeply revised by the authors. The first paragraph of the discussion could be removed, since it talks about pathophysiology of the disease, this is not a discussion.

Response: We agree with the reviewer. The paragraph of the discussion has additionally been revised as suggested

Comment #6

There are many data in the discussion section that were provided in table 2 and should be not be duplicated.

Response:  We apologize. We agree with the reviewer.  Thus, we included Table of AKI risk factors in discussion. We revised discussion to minimize the redundancy.

Comment #7

Authors should analyses their results, compare the included studies and try to explain, for example, the huge variability between the analsysed studies regarding the development of AKI, risk factors, possible causes for these differences.

Response:  We appreciated the reviewer’s input. The variability of AKI and its outcome among included studies are likely due to difference in year of study, country, patient population, clinical practice, and definition of AKI and mortality outcomes. Table 1 described the main characteristics of included studies. Meta-regression was perform to assess the impact of these factors on AKI incidence.

We greatly appreciated the reviewer’s time and comments to improve our manuscript.

Round 2

Reviewer 2 Report

The authors have correctly improved the manuscript.